# Treatment Outcomes and Risk Factors for Incomplete Treatment after Definitive Chemoradiotherapy for Non-Resectable or Metastatic Esophageal Cancer

**DOI:** 10.3390/cancers15225421

**Published:** 2023-11-15

**Authors:** Chu-Pin Pai, Ling-I Chien, Chien-Sheng Huang, Han-Shui Hsu, Po-Kuei Hsu

**Affiliations:** 1Division of Thoracic Surgery, Department of Surgery, Lotung Poh-Ai Hospital, Ilan 26546, Taiwan; bcbjohnny@gmail.com; 2School of Medicine, National Yang Ming Chiao Tung University, Taipei 30010, Taiwan; huangcs@vghtpe.gov.tw (C.-S.H.); hsuhs@vghtpe.gov.tw (H.-S.H.); 3Department of Nursing, Taipei Veterans General Hospital, Taipei 112201, Taiwan; lichien@vghtpe.gov.tw; 4Division of Thoracic Surgery, Department of Surgery, Taipei Veterans General Hospital, Taipei 112201, Taiwan

**Keywords:** esophageal cancer, esophageal squamous cell carcinoma, definitive, chemoradiotherapy, adverse events, incomplete treatment

## Abstract

**Simple Summary:**

The rates of adverse events and incomplete treatment remain high among patients with unresectable or metastatic esophageal cancer who receive definitive treatment. The overall survival and treatment-related adverse events were analyzed. Subgroup analysis was performed based on the completeness of the treatment plan. Complete treatment was positively correlated with increased survival. In multivariable analysis, poor performance, bone metastasis, airway invasion, and liver cirrhosis are risk factors for incomplete treatment.

**Abstract:**

Among patients with unresectable or metastatic esophageal cancer who receive definitive chemotherapy or chemoradiotherapy, the rates of treatment-related adverse events and incomplete treatment remain high. We conducted this study to investigate survival after definitive treatments and identify predicting factors for incomplete treatment. The data of patients who received definitive chemotherapy or chemoradiotherapy for esophageal cancer were retrospectively examined. The patients were assigned to Group 1: incomplete definitive treatment; Group 2: complete definitive treatment; or Group 3: complete definitive treatment with additional salvage surgery. The data of 273 patients (90, 166, and 17 in Groups 1, 2, and 3, respectively) were analyzed. In the survival analysis, the median overall survival of Groups 1, 2, and 3 were 2.6, 10.3, and 29.5 months, respectively. A significant difference in 3-year overall survival was observed among the groups (2.2%, 12.4%, and 48.5%, *p* < 0.001). In multivariable analysis, the independent risk factors for incomplete definitive treatment included poor performance score (hazard ratio (HR): 5.23, *p* = 0.001), bone metastasis (HR: 2.18, *p* = 0.024), airway invasion (HR: 2.90, *p* = 0.001), and liver cirrhosis (HR: 3.20, *p* = 0.026). Incomplete definitive treatment is associated with a far worse prognosis. Poor performance, bone metastasis, airway invasion, and liver cirrhosis are risk factors for incomplete treatment.

## 1. Introduction

Esophageal cancer is the sixth most common cause of cancer-related death for men and the ninth most common cause for women worldwide. Surgical resection of esophageal tumors is generally considered the cornerstone of curative approaches in localized esophageal cancer. However, for patients with locally advanced non-resectable or metastatic esophageal cancer, surgical resection is not a viable option and definitive chemotherapy or chemoradiotherapy is considered a standard treatment [1,2,3]. Several investigators have, therefore, attempted to evaluate the outcomes of definitive chemoradiation therapy. The efficacy and safety of a definitive chemoradiotherapy regimen have been confirmed in previous trials. Despite the survival benefits, the rates of regimen-related adverse events, including life-threatening toxicities and toxicity-related deaths, remain high. As many as 41% of patients cannot complete their planned chemoradiotherapy regimen [2,4,5,6]. The survival for this group of non-surgical candidates remains poor, and incomplete definitive therapy leads to worse prognoses.

To better understand the treatment response and limitations of definitive chemoradiation therapy, we conducted this retrospective study to investigate the outcome, survival prognosticators, and risk factors for incomplete therapy among patients with locally advanced non-resectable or metastatic esophageal cancer.

## 2. Materials and Methods

The study protocol was reviewed and approved by the Institutional Review Board of Taipei Veterans General Hospital (TPEVGH2015-06-001BC).

### 2.1. Study Population

A prospectively maintained database was queried for patients with esophageal malignancies between January 2010 and December 2019 at TPEVGH. The inclusion criteria were patients who received definitive chemotherapy or chemoradiotherapy as the initial treatment plan. Patients with non-metastatic and resectable disease were excluded.

The staging workup included a systemic physical examination, standard laboratory screening, esophagogastroscopy (endoscopic ultrasound; EUS), bronchoscopy for tumors in the upper or middle third of the esophagus, computed tomography (CT) scanning from the neck to the upper abdomen, and whole-body fluorodeoxyglucose positron emission tomography/CT (FDG PET/CT). A multidisciplinary team conference was present, which comprised surgeons, medical and radiation oncologists, gastroenterologists, pathologists, radiologists, and special nurses, for discussion and recommendations regarding the treatment plan and its modification.

### 2.2. Staging Evaluations

All patients were staged according to the American Joint Committee on Cancer staging criteria, 7th edition [7]. Adjacent organ invasion was evaluated using esophageal EUS, bronchoscopy, and/or CT. Invasion was diagnosed if the bronchoscopy showed protrusion of the esophageal tumor into the trachea and/or bronchi or abnormal tracheal mucosa, or if EUS showed invasion of the trachea, bronchus, aorta, and/or other peripheral organs. If the patient could not undergo EUS, adjacent organ invasion was defined using CT or PET-CT. Generally, invasion of the airway was diagnosed based on the loss of fat planes between the esophagus and the trachea/bronchus, combined with simultaneous compressive deformation of the trachea/bronchus or the tumor directly protruding into the trachea/bronchus. Invasion of the aorta was defined as >90° of the aorta being surrounded by tumor in more than one CT slice.

The comorbidities of the study population were documented within the database, and diagnoses were made based on laboratory and image findings. In particular, diabetes mellitus was diagnosed based on the World Health Organization and International Diabetes Federation diagnostic criteria. Coronary artery disease was identified using coronary angiography. Chronic obstructive pulmonary disease was confirmed using a spirometry test of pulmonary function. Liver cirrhosis was diagnosed based on laboratory data, endoscopic findings, and sonography or CT scans.

### 2.3. Definitive Chemoradiotherapy and Surgery

Definitive chemoradiotherapy comprised a combination of fluoropyrimidine and platinum or taxane with concurrent external beam radiotherapy (cumulative dose of 50.4 Gy or above, in fractions of 1.8 Gy). Tumor response was defined based on the Response Evaluation Criteria in Solid Tumor, version 1.1 [8]. Toxicity was graded according to the Common Terminology Criteria for Adverse Events, version 5.0 [9]. Completion of definitive treatment was defined as more than 80% of planned chemotherapy and more than 90% of the radiotherapy dose. Patients who did not finish the complete treatment course of definitive chemoradiotherapy were assigned to Group 1. The clinical conditions and reason for incompletion were analyzed. For those who completed definitive chemoradiotherapy, the decision between further treatment or close surveillance was based on the tumor response, performance status of the patient, and curability of salvage esophagectomy. An organ-sparing strategy with close monitoring was discussed with patients who demonstrated clinical complete response. Patients with failed definitive chemoradiotherapy, exhibiting either residual disease or relapse during follow-up, were candidates for salvage esophagectomy if curative resection was possible and tolerable. Patients who completed only definitive chemoradiotherapy were assigned to Group 2, whereas those with additional salvage esophagectomy were assigned to Group 3.

The details of surgery at our institution were as previously described [10]. After the operation or initiation of definitive chemotherapy, follow-up evaluations were arranged, including clinical and laboratory testing as well as chest CT every 3–4 months for the first 2 years, every 6 months between the second and fifth years, and every year after the fifth year.

### 2.4. Statistics

Continuous variables were either recorded as means and compared using Student’s *t*-test or summarized as medians and compared using the Mann–Whitney *U* test. Categorical variables were recorded as absolute counts and compared using the chi-square test or Fisher’s exact test. Overall survival (OS) was defined as the time from the beginning of radiotherapy until death or the last known follow-up, based on either medical records or a follow-up phone call. Follow-up time was defined as the time from the beginning of initial treatment until death or the last known follow-up, based on either medical records or a follow-up phone call. Survival curves were plotted using the Kaplan–Meier method and compared using the log-rank test. Univariable and multivariable Cox regression modeling was used to identify prognostic factors [11]. Factors with a *p* value < 0.05 in univariable analysis were included in multivariable modeling. All statistical analyses were conducted using Statistical Product and Service Solutions, version 25 (IBM Corp, Armonk, NY, USA), and a two-sided *p* value < 0.05 was considered statistically significant.

## 3. Results

### 3.1. Clinicopathological Characteristics of the Study Patients

Between January 2010 and December 2019, 532 patients received definitive treatment for esophageal cancer in our hospital. After the exclusion criteria were applied, the data of the remaining 273 patients with advanced non-resectable or metastatic disease were analyzed (Figure 1).

The clinical and pathological characteristics of the patients are summarized in Table 1. Among the 273 patients, 90 (33.0%) patients did not complete the treatment course (Group 1). Of the remaining 183 patients, 166 completed definitive treatment as planned (Group 2), and 17 patients underwent salvage surgical treatment after definitive chemotherapy or chemoradiotherapy (Group 3). Among the patients in Groups 2 and 3, the performance status and laboratory profile were better, and the clinical T staging of the disease was lower compared with the patients in Group 1. Notably, multidisciplinary meetings throughout the treatment course were more frequent in Groups 2 and 3. Regarding the treatment regimen, cisplatin plus 5-fluouracil was the most used combination. Other regimens included single platinum in three patients in Group 1, single 5FU in three patients in Group 1, and three patients in Group 2. Taxane-based combination treatment was used in four patients in Group 1 and 12 patients in Group 2. Platinum combination with immunotherapy was used in two patients in Group 2 (nivolumab was used for one, and pembrolizumab was used for the other).

### 3.2. Survival and Prognostic Factors for Patients Receiving Surgical and Non-Surgical Treatment

In the survival analysis, the median follow-up time for all patients was 8.1 months (interquartile range: 3.9–16.1). The 1- and 3-year OS rates in the entire cohort were 33.9% and 11.2%, respectively. The 1- and 3-year OS rates were 3.3% and 2.2% in Group 1, 45.3% and 12.4% in Group 2, and 87.5% and 48.5% in Group 3, respectively (Figure 2).

A Cox proportional hazards regression model was used to analyze prognostic factors for OS in the entire cohort (Table 2). The significant prognostic factors in univariable analysis for OS included sex, performance status, level of serum albumin, serum neutrophil-to-lymphocyte ratio, clinical T and N stage, liver metastasis, bone metastasis, and clinical complete response. Among these factors, clinical N3 stage (hazard ratio [HR]: 1.48, 95% confidence interval [CI]: 1.14–1.93, *p* = 0.004), clinical T4b stage of airway involvement (HR: 1.95, 95% CI: 1.42–2.67, *p* < 0.001), clinical M1 stage of liver metastasis (HR: 1.78, 95% CI: 1.27–2.48, *p* = 0.001), clinical M1 stage of bone metastasis (HR: 1.73, 95% CI: 1.23–2.43, *p* = 0.002), and clinical complete response (HR: 0.05, 95% CI: 0.02–1.17, *p* < 0.001) remained as independent prognostic factors in the multivariable analysis.

### 3.3. Risk Factors for Incomplete Definitive Chemoradiotherapy

Table 3 shows the results of univariable and multivariable analysis for risk factors for incomplete planned definitive therapy. Performance status, pre-treatment level of serum hemoglobin, albumin, neutrophil-to-lymphocyte ratio, lymphocyte-to-monocyte ratio, clinical T stage, clinical T4b stage of airway involvement, clinical M1 stage of bone metastasis, liver cirrhosis, and frequency of multidisciplinary team conference were significantly associated with incomplete treatment in univariable analysis. After multivariable analysis, performance status with an ECOG score ≥ 2 (HR: 5.23, 95% CI: 1.95–14.02, *p* = 0.001), clinical T4b stage of airway involvement (HR: 2.90, 95% CI: 1.53–5.51, *p* = 0.001), liver cirrhosis (HR: 3.20, 95% CI: 1.15–8.91, *p* = 0.026), and clinical M1 stage of bone metastasis (HR: 2.18, 95% CI: 1.11–4.30, *p* = 0.024) were significant independent risk factors.

Table 4 shows adverse events of grade 3 or higher in the entire cohort, including leukopenia in 101 (37%) patients, anemia in 60 (22%), thrombocytopenia in 50 (18%), respiratory system events in 64 (24%), gastrointestinal tract events in 44 (16%), and renal insufficiency in 5 (2%).

## 4. Discussion

### 4.1. Survival and Prognosticators of Patients Receiving Definitive Chemoradiotherapy

In our study, the survival outcome was poor, and the 3-year OS of 11.2% was comparable to that in previous studies, which ranged from 0% to 27% [3,12,13,14,15]. The survival of Group 1 patients in this study was the worst and was similar to those who received optimal supportive care in previous studies [15,16]. Group 3 patients exhibited the longest survival, which was compatible with previous findings [15,17,18].

The regimen of the chemotherapy used in this study was mainly platin based, which was also used in previous clinical trials. A standard-dose platin-based regimen is considered superior to a low-dose regimen and is routinely used in our hospital [14,19]. On the other hand, a tegafur-based regimen has been shown to be well tolerated with acceptable outcomes in previous studies and was used in 2.6% of our patients [20]. Moreover, similar survival outcomes between regimens of oxaliplatin with 5-FU (FOLFOX) and cisplatin with 5-FU were reported in a previous trial, and this regimen was used for approximately 1% of our patients [21].

Based on the results of the previous RTOG9405 trial, a standard dose of 50.4 Gy is used for advanced esophageal cancer because of excessive treatment-related deaths at higher radiation doses [3]. However, most of the deaths (7/11) occurred before a cumulative dose of 50 Gy was delivered, and only one death was directly attributable to high-dose radiation. Hence, an irradiation dose of 60 Gy has been used in JCOG clinical trials as well as in our hospital. According to the results of the recent CONCORDE and ARTDECO studies, the escalation of radiation doses above 50.4 Gy for primary tumors does not result in a significant increase in local control [22,23]. The SCOPE 2 trial assessed whether a dose effect could be identified [24].

The aforementioned ARTDECO study recruited patients with a mean age of over 70 years and compared the treatment outcomes of standard-dose (SD) and high-dose (HD) radiation groups. This study used a higher daily radiation dose in the HD group (2.2 Gy) than in the SD group (1.8 Gy), which could have led to more radiation reactions. The analysis of overall adverse events demonstrated grade 4 and 5 common toxicity criteria of 12% and 5% in the SD group versus 14% and 10% in the HD group, respectively (*p* = 0.15). Although the difference was not significant, more grade 4 and 5 common toxicities were observed. In our study, individual grade 3 or 4 toxicities generally occurred at similar rates as in previous trials. Anemia and adverse events of the gastrointestinal tract and respiratory system occurred more frequently in Group 1 patients. The rate of treatment completion ranged from 54% to 76% in previous trials, and similarly, the completion rate in our study was 67% (183/273) [5,21,25]. Improved nutritional support, better control of emesis, and the differing study design all contributed to the completion rate.

Increased alcohol consumption is a common risk factor for esophageal cancer and liver cirrhosis, and approximately 3–14% of patients with esophageal cancer are reported to have liver cirrhosis [26]. Cirrhosis slows the liver’s ability to process nutrients, hormones, drugs, and natural toxins and results in immune system dysfunction, malnutrition, fluid imbalance, and bleeding diathesis [27]. Published studies have shown that cirrhosis poses major therapeutic challenges for esophagectomy [26]. Medical treatment is often the main option for these patients, and therefore, the tolerability of definitive treatment is critical. However, little is known about the treatment outcomes of cirrhotic patients. One case–control study suggested the less aggressive treatment in advanced liver cirrhosis [28]. In our study, liver cirrhosis was associated with an increased risk of incomplete treatment. For this group of patients, previous studies have introduced alternatives to current first-line chemotherapy, such as immune therapy or oral anticancer drugs that offer satisfying survival and tumor control with a lower or equal rate of adverse events. On the other hand, we also identified a clinical T4b stage of airway involvement as an independent risk factor for incomplete treatment. Of these patients, 49% (31/63) could not complete the treatment course. A significantly higher treatment completion rate of 90% was identified in previous studies of patients with cT4 disease [29,30]. The difference may be due to the fact that the CDDP plus 5-FU regimen was used in nearly 90% of our study patients, compared to 14% in previous studies. Moreover, in our study, treatment completion was defined as 80% or more of the chemotherapy completed and 90% or more of the radiotherapy dose completed. The strict criteria may also have contributed to the difference in the completion rate. Nevertheless, another study reported that the tolerability of chemoradiotherapy for cT4b patients was limited, and the rate of fistula formation was 30.1% [31]. A comprehensive evaluation of the risk based on clinical characteristics should be conducted to select the appropriate candidates for chemoradiotherapy, and the regimen should be modified as soon as signs of intolerance appear.

### 4.2. Prognostic Effect of Salvage Esophagectomy

The 3-year OS ranged from 14% to 54% in previous studies of salvage esophagectomy after definitive chemoradiotherapy [32]. The 3-year OS in Group 3 patients in this study was 48.5%, which was comparable to that of previous studies. Recent investigations have suggested that approximately a quarter of patients exhibit disease recurrence after a clinical complete response, and salvage esophagectomy is superior in outcomes and complications compared to salvage chemoradiotherapy [33,34,35]. To achieve long-term survival, R0 resection is essential regardless of the response to definitive chemoradiation therapy, and patients with upper thoracic esophageal tumors are at risk of incomplete resection [16,36,37,38,39,40]. Despite the low percentage of these patients completing treatment, salvage surgery can achieve favorable long-term survival outcomes.

### 4.3. Optimization of Treatment Plans

In the setting of definitive chemoradiotherapy, not all patients could complete the planned treatment course, and we were able to identify significant risk factors for incomplete treatment. No previous study has demonstrated the pre-treatment risk factors for incompletion, and we believe this study could serve as a guide for risk stratification and help optimize treatment planning.

Certain limitations of this study should be acknowledged. Firstly, the diagnosis of cT4b was mainly based on CT imaging and bronchoscopy findings. Not all patients can undergo EUS, which provides detailed information for adjacent organ invasion. Secondly, the number of cirrhotic patients of Child–Pugh class B was relatively small, and therefore, we could not perform further risk stratification based on cirrhotic classes. Additionally, 90 patients in the cohort could not complete the treatment, and therefore, survival endpoints such as disease-free survival could not be fully evaluated in this study. Finally, the chemotherapy regimen of the study patients mainly comprised CDDP plus 5-FU, and thus, we were unable to perform a comprehensive comparison of regimens.

### 4.4. Brief Summary

Data from 273 patients between 2010 and 2019 with advanced non-resectable or metastatic disease were analyzed. Among the 273 patients, 90 patients did not complete the treatment course (Group 1). Of the remaining 183 patients, 166 completed definitive treatment as planned (Group 2), and 17 patients underwent salvage surgical treatment after definitive chemotherapy or chemoradiotherapy (Group 3). Group 3 had the best OS, and Group 1 had the worst OS. Clinical N3 stage, airway involvement, liver metastasis, bone metastasis, and clinical complete response were independent prognostic factors. In the analysis for incomplete treatment, an ECOG score ≥ 2, airway involvement, liver cirrhosis, and bone metastasis were significant independent risk factors. Adverse events in the respiratory system and gastrointestinal tract were higher among Group 1 patients.

## 5. Conclusions

Despite the acceptable outcomes associated with definitive treatment for non-resectable or metastatic esophageal cancer, the accompanying adverse events result in limitations. The increased risk of incomplete treatment in patients with cirrhosis or airway involvement indicates that a prior comprehensive evaluation of the risk and timely regimen modification should be conducted to optimize the treatment plan and prolong survival.

## Figures and Tables

**Figure 1 cancers-15-05421-f001:**
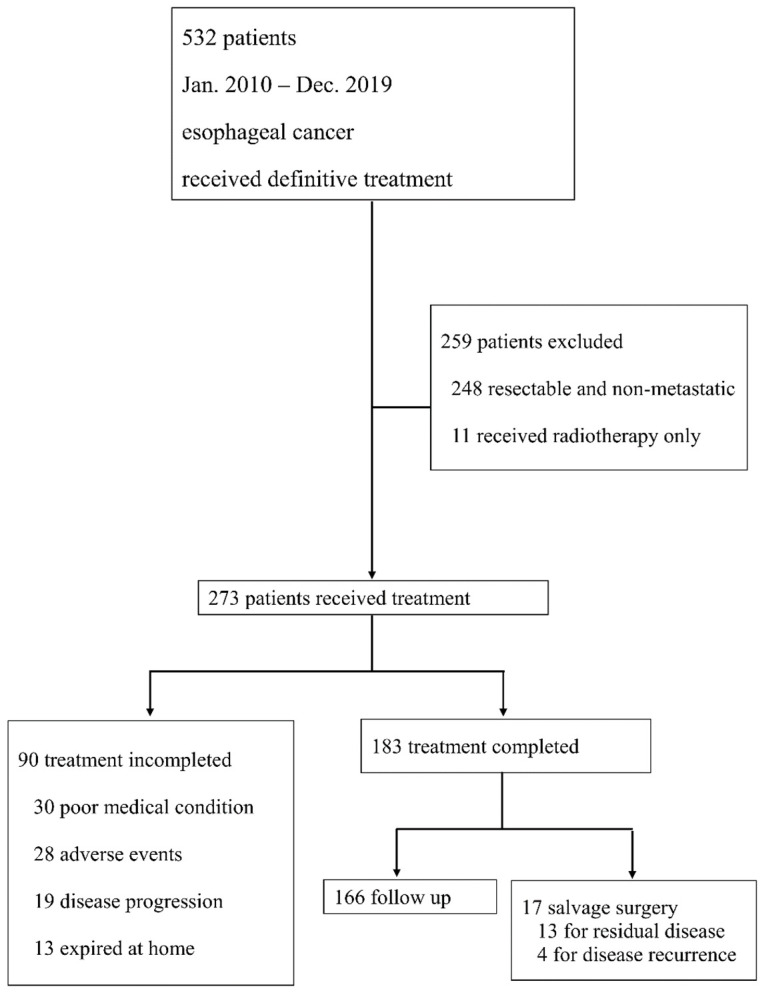
Flow diagram of the selection of patients in this study.

**Figure 2 cancers-15-05421-f002:**
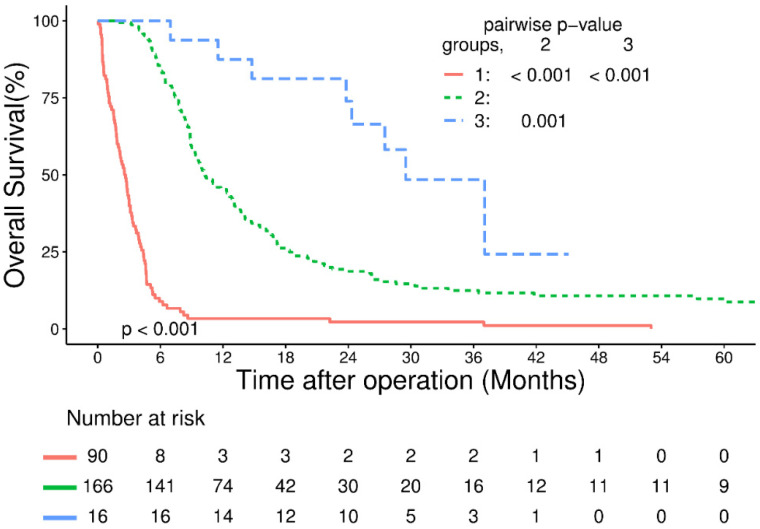
Overall survival curves categorized by patient groups. Group 1: patients who did not complete definitive therapy. Group 2: patients who completed definitive therapy without surgery. Group 3: patients who completed definitive therapy and received salvage esophagectomy.

**Table 1 cancers-15-05421-t001:** Clinical characteristics of patients categorized by completion of definitive treatment and salvage operation.

	All Patients (N = 273)		dCRT Incomplete (N = 90)	dCRT Complete (N = 183)			
		*p* *	Total	Total	Salvage Operation		*p* **
			Group 1		No (N = 166) Group 2	Yes (N = 17) Group 3	
Age (years; median, IQR)	59.0 (52.0–67.0)	0.282	61.0 (52.0–68.0)	58.0 (52.0–66.0)	58.0 (51.8–66.3)	60.0 (54.0–66.5)	0.886
Sex (%)		0.287					0.611
Male	251 (91.9)		85 (94.4)	166 (90.7)	150 (90.4)	16 (94.1)	
Female	22 (8.1)		5 (5.6)	17 (9.3)	16 (9.6)	1 (5.9)	
BMI (median, IQR)	21.0 (19.0–23.7)	0.087	20.4 (17.9–23.2)	21.2 (19.3–23.8)	21.1 (19.2–23.8)	23.2 (20.3–25.2)	0.091
Smoking (%)		0.065					0.883
Never	53 (20.2)		23 (25.8)	30 (16.4)	27 (16.3)	3 (17.6)	
Former/current	219 (79.6)		66 (74.2)	153 (83.6)	139 (83.7)	14 (82.4)	
ECOG (%)		<0.001					0.580
0	142 (52.0)		29 (32.2)	113 (61.7)	101 (60.8)	12 (70.6)	
1	106 (38.8)		43 (47.8)	63 (34.4)	58 (34.9)	5 (29.4)	
≥2	25 (9.2)		18 (20.0)	7 (3.8)	7 (4.2)	0 (0)	
Cirrhosis (%)		0.007					0.355
Child–Pugh A			10 (11.2)	8 (4.4)	8 (4.8)	0 (0)	
Child–Pugh B			2 (2.2)	0 (0)	0 (0)	0 (0)	
Charlson comorbidity index	3.0 (2.0–4.0)	0.131	3.0 (2.0–4.3)	3.0 (2.0–4.0)	3.0 (2.0–4.0)	2.0 (2.0–3.5)	0.703
Laboratory data (median, IQR)							
Hemoglobin (g/dL)	12.4 (10.9–13.6)	0.003	11.9 (10.1–13.3)	12.6 (11.3–13.8)	12.6 (11.2–13.8)	13.5 (12.3–14.1)	0.150
Albumin (g/dL)	3.7 (3.3–4.0)	<0.001	3.5 (3.0–3.9)	3.8 (3.4–4.1)	3.8 (3.4–4.0)	3.9 (3.6–4.3)	0.017
Neutrophil (cells/µL)	5427.1 (4126.0–7392.6)	0.109	5505.2 (4203.7–8296.7)	5217.2 (4002.3–7145.7)	5544.0 (4301.5–7204.4)	3938.7 (2989.2–5036.8)	0.018
Lymphocyte (cells/µL)	1504.4 (1124.3–1850.0)	0.009	1354.2 (1024.6–1731.3)	1572.6 (1163.6–1894.8)	1548.6 (1153.1–1862.3)	1674.8 (1405.9–2154.4)	0.136
Monocyte (cells/µL)	633.2 (490.2–861.6)	0.509	636.0 (463.3–910.6)	629.9 (501.0–846.3)	633.9 (520.4–859.5)	555.0 (458.3–763.4)	0.692
Platelet (counts/µL)	279,000.0 (217,000.0–342,000.0)	0.191	285,000.0 (186,500.0–346,000.0)	277,000.0 (227,000.0–341,000.0)	279,000.0 (228,000.0–345,000.0)	248,000.0 (222,000.0–383,000.0)	0.081
NLR (IQR)	3.7 (2.6–5.5)	0.001	4.5 (2.9–7.2)	3.5 (2.6–4.9)	3.6 (2.7–5.0)	2.2 (1.7–3.4)	0.002
LMR (IQR)	2.2 (1.6–3.0)	0.009	1.9 (1.4–2.8)	2.4 (1.7–3.2)	2.2 (1.7–3.1)	3.5 (2.3–3.8)	0.012
PLR (IQR)	184.1 (134.2–255.4)	0.376	190.9 (131.2–277.8)	178.3 (135.6–238.9)	184.5 (139.6–252.0)	147.8 (100.2–182.4)	0.008
Tumor location (%)		0.610					0.867
Cervical	24 (8.8)		8 (8.9)	16 (8.7)	15 (9.0)	1 (5.9)	
Upper	69 (25.3)		26 (28.9)	43 (23.5)	40 (24.1)	3 (17.6)	
Middle	99 (36.3)		32 (35.6)	67 (36.6)	59 (35.5)	8 (47.1)	
Lower	72 (26.4)		23 (25.6)	49 (26.8)	45 (27.1)	4 (23.5)	
EGJ	9 (3.3)		1 (1.1)	8 (4.4)	7 (4.2)	1 (5.9)	
cT stage (%)		0.004					0.758
1	5 (1.8)		0 (0)	5 (2.7)	4 (2.4)	1 (5.9)	
2	30 (11.0)		8 (8.9)	22 (12.0)	20 (12.0)	2 (11.8)	
3	146 (53.5)		39 (43.3)	107 (58.5)	96 (57.8)	11 (64.7)	
4	87 (31.9)		42 (46.7)	45 (24.6)	42 (25.3)	3 (17.6)	
Cannot be assessed	5 (1.8)		1 (1.1)	4 (2.2)	4 (2.4)	0 (0)	
cN stage (%)		0.648					0.511
0	17 (6.2)		6 (6.7)	11 (6.0)	11 (6.6)	0 (0)	
1	61 (22.3)		20 (22.2)	41 (22.4)	38 (22.9)	3 (17.6)	
2	74 (27.1)		26 (28.9)	48 (26.2)	41 (24.7)	7 (41.2)	
3	120 (44.0)		37 (41.1)	83 (45.4)	76 (45.8)	7 (41.2)	
Cannot be assessed	1 (0.4)		1 (1.1)	0 (0.0)	0 (0.0)	0 (0.0)	
cM stage (%)		0.567					0.766
0	67 (24.5)		24 (26.7)	43 (23.5)	40 (24.1)	3 (17.6)	
1	206 (75.5)		66 (73.3)	140 (76.5)	126 (75.9)	14 (82.4)	
Histology (%)		0.621					0.715
SCC	249 (91.2)		81 (90.0)	168 (91.8)	152 (91.6)	16 (94.1)	
ADC	24 (8.8)		9 (10.0)	15 (8.2)	14 (8.4)	1 (5.9)	
MDT conference (%)		<0.001					<0.001
0	109 (39.9)		46 (51.1)	63 (34.4)	62 (37.3)	1 (5.9)	
1	112 (41.0)		39 (43.3)	73 (39.9)	70 (42.2)	3 (17.6)	
≥2	52 (19.1)		5 (5.6)	47 (25.7)	34 (20.5)	13 (76.5)	
Radiation dose (cGy) (%)							
≤5040	125 (45.8)	<0.001	59 (65.6)	66 (36.1)	59 (35.5)	7 (41.2)	0.792
>5040	148 (54.2)		31 (34.4)	117 (63.9)	107 (64.5)	10 (58.8)	
Chemotherapy regimen (%)		0.192					0.800
CDDP + 5-FU	230 (84.2)		77 (85.6)	153 (83.6)	138 (83.1)	15 (88.2)	
CDDP + 5-FU + Taxane	10 (3.7)		1 (1.1)	9 (4.9)	9 (5.4)	0 (0)	
Tegafur–Uracil	7 (2.6)		1 (1.1)	6 (3.3)	5 (3.0)	1 (5.9)	
5-FU	6 (2.2)		3 (3.3)	3 (1.6)	3 (1.8)	0 (0)	
Carboplatin + Taxane	4 (1.5)		3 (3.3)	1 (0.5)	1 (0.6)	0 (0)	
Others ***	16 (5.9)		5 (5.6)	11 (6.0)	10 (6.0)	1 (5.9)	
Clinical response to therapy (%)		<0.001					<0.001
CR	16 (5.8)		0 (0)	16 (8.7)	11 (6.6)	5 (29.4)	
PR	112 (41.0)		58 (64.4)	54 (29.5)	43 (25.9)	11 (64.7)	
SD	23 (8.4)		13 (14.4)	10 (5.5)	10 (6.0)	0 (0)	
PD	119 (43.6)		19 (21.1)	100 (54.6)	99 (59.6)	1 (5.9)	
Undefined	3 (1.1)		0 (0)	3 (1.6)	3 (1.8)	0 (0)	
Follow-up time (month; median, IQR)	8.1 (3.9–16.1)	<0.001	2.6 (1.1–4.4)	11.4 (7.7–21.7)	10.2 (7.4–18.2)	26.5 (18.5–33.3)	0.071

* *p* value for dCRT complete group and dCRT incomplete group; ** *p* value for Groups 2 and 3; *** other regimens included Xeloda monotherapy, CDDP monotherapy, carboplatin plus 5-FU, CDDP plus taxane, oxaliplatin plus 5-FU in 2, CFHx, FLOT, and chemotherapy combined immunotherapy. Abbreviations: IQR: interquartile range; BMI: body mass index; ECOG: Eastern Cooperative Oncology Group; NLR: neutrophil-to-lymphocyte ratio; LMR: lymphocyte-to-monocyte ratio; PLR: platelet-to-lymphocyte ratio; EGJ: esophagogastric junction; SCC: squamous cell carcinoma; ADC: adenocarcinoma; MDT: multidisciplinary team; CDDP: cis-diamminedichloroplatinum; 5-FU: 5-fluouracil; CFHx: cisplatin plus 5-FU plus hydroxyurea; FLOT: 5-FU plus leucovorin plus oxaliplatin plus docetaxel; CR: complete response; PR: partial response; SD: stable disease; PD: progressive disease.

**Table 2 cancers-15-05421-t002:** Univariable and multivariable analysis for OS in patients who received dCRT or definitive chemotherapy.

Variable	Univariate			Multivariate		
	HR	95% CI	*p* Value	HR	95% CI	*p* Value
Age (years; ≥60 vs. <60)	0.84	0.65–1.08	0.181			
Sex (male vs. female)	1.69	1.04–2.74	0.033	1.70	0.99–2.91	0.051
Smoking (ever vs. never)	0.77	0.56–1.05	0.102			
BMI (≥21 vs. <21)	0.94	0.73–1.21	0.630			
ECOG (≥2 vs. 0–1)	2.13	1.40–3.23	<0.001	1.58	0.95–2.65	0.081
CCI score (≥4 vs. <4)	1.04	0.80–1.35	0.777			
Liver cirrhosis (yes vs. no)	1.20	0.75–1.93	0.452			
Laboratory data						
Hemoglobin (g/dL; ≥10 vs. <10)	0.80	0.57–1.11	0.180			
Albumin (g/dL; ≥3.5 vs. <3.5)	0.69	0.54–0.90	0.005	0.85	0.63–1.14	0.273
NLR (≥3.4 vs. <3.4)	1.41	1.09–1.82	0.009	1.02 1.23	0.98–1.06 0.93–1.63	0.295 0.152
LMR (<2.4 vs. ≥2.4)	1.23	0.96–1.59	0.109			
PLR (≥170 vs. <170)	1.23	0.96–1.59	0.106			
Tumor location (Ce vs. Ut/Mt/Lt/EGJ)	0.94	0.60–1.47	0.776			
c T stage (4b vs. 1/2/3/4a)	1.48	1.12–1.96	0.005			
c N stage (3 vs. 0/1/2)	1.35	1.05–1.75	0.020	1.38	1.06–1.80	0.019
c M stage (1 vs. 0)	0.83	0.62–1.10	0.190			
cT4b						
Aorta	1.41	0.88–2.25	0.152			
Airway	1.65	1.23–2.21	0.001	1.63	1.17–2.25	0.003
Distant metastasis						
Retroperitoneal LN	1.24	0.93–1.65	0.149			
Liver	1.38	1.01–1.90	0.045	1.66	1.18–2.33	0.004
Bone	1.75	1.29–2.38	<0.001	1.71	1.23–2.38	0.002
Lung	0.94	0.72–1.23	0.647			
Adrenal	1.42	0.73–2.76	0.308			
Histopathology (SCC vs. ADC)	1.24	0.80–1.93	0.341			
MDT (yes vs. no)	0.82	0.63–1.05	0.117			
Clinical response						
PD	1			1		
PR	0.80	0.61–1.06	0.120	0.82	0.61–1.10	0.190
SD	1.06	0.67–1.67	0.807	1.05	0.60–1.83	0.863
CR	0.05	0.01–0.16	<0.001	0.05	0.02–0.17	<0.001

Abbreviations: OS: overall survival; dCRT: definitive chemoradiotherapy; HR: hazard ratio; CI: confidence interval; BMI: body mass index; ECOG: Eastern Cooperative Oncology Group; CCI: Charlson comorbidity index; NLR: neutrophil-to-lymphocyte ratio; LMR: lymphocyte-to-monocyte ratio; PLR: platelet-to-lymphocyte ratio; EGJ: esophagogastric junction; SCC: squamous cell carcinoma; ADC: adenocarcinoma; MDT: multidisciplinary team; CR: complete response; PR: partial response; SD: stable disease; PD: progressive disease.

**Table 3 cancers-15-05421-t003:** Univariable and multivariable analysis for incomplete dCRT or definitive chemotherapy.

Variable	Univariate			Multivariate		
	HR	95% CI	*p* Value	HR	95% CI	*p* Value
Age (years; ≥60 vs. <60)	1.51	0.91–2.50	0.114			
Sex (male vs. female)	1.74	0.62–4.88	0.292			
Smoking (ever vs. never)	0.56	0.30–1.04	0.067			
BMI (≥21 vs. <21)	0.76	0.46–1.27	0.299			
ECOG (≥2 vs. 0–1)	6.29	2.52–15.70	< 0.001	5.23	1.95–14.02	0.001
CCI score (≥4 vs. <4)	1.55	0.92–2.63	0.102			
Laboratory data						
Hemoglobin (g/dL; ≥10 vs. <10)	0.44	0.23–0.84	0.014	0.64	0.31–1.34	0.236
Albumin (g/dL; ≥3.5 vs. <3.5)	0.49	0.29–0.82	0.006			
NLR (≥3.4 vs. <3.4)	1.78	1.05–3.03	0.033	1.06	0.97–1.17	0.202
LMR (<2.4 vs. ≥2.4)	1.77	1.04–3.00	0.034	1.01	0.73–1.37	0.993
PLR (≥170 vs. <170)	1.44	0.85–2.42	0.174			
Tumor location (Ce vs. Ut/Mt/Lt/EGJ)	1.02	0.42–2.48	0.968			
c T stage (4b vs. 1/2/3/4a)	2.08	1.20–3.61	0.009			
c N stage (3 vs. 0/1/2)	0.86	0.51–1.43	0.556			
c M stage (1 vs. 0)	0.85	0.47–1.51	0.568			
cT4b						
Aorta	1.58	0.64–3.91	0.319			
Airway	2.48	1.39–4.42	0.002	2.90	1.53–5.51	0.001
Distant metastasis						
Retroperitoneal LN	1.25	0.71–2.22	0.443			
Liver	1.72	0.92–3.20	0.089			
Bone	2.40	1.32–4.36	0.004	2.18	1.11–4.30	0.024
Lung	1.08	0.63–1.86	0.786			
Adrenal	1.37	0.38–4.99	0.631			
Histopathology (SCC vs. ADC)	0.80	0.34–1.91	0.621			
MDT (yes vs. no)	0.50	0.30–0.84	0.009	0.65	0.37–1.15	0.139
Liver cirrhosis (yes vs. no)	3.41	1.34–8.67	0.010	3.20	1.15–8.91	0.026

Abbreviations: dCRT: definitive chemoradiotherapy; HR: hazard ratio; CI: confidence interval; BMI: body mass index; ECOG: Eastern Cooperative Oncology Group; CCI: Charlson comorbidity index; NLR: neutrophil-to-lymphocyte ratio; LMR: lymphocyte-to-monocyte ratio; PLR: platelet-to-lymphocyte ratio; SCC: squamous cell carcinoma; EGJ: esophagogastric junction; ADC: adenocarcinoma; MDT: multidisciplinary team.

**Table 4 cancers-15-05421-t004:** Adverse events in patients who received definitive chemoradiotherapy or chemotherapy.

	dCRT Incomplete (N = 90)		dCRT Complete (N = 183)	
	Grade 3	Grade 4	Grade 3	Grade 4
Hematological (%)				
Leukopenia	15 (16.7)	18 (20.0)	48 (26.2)	20 (10.9)
Anemia	23 (25.6)	2 (2.2)	33 (18.0)	2 (1.1)
Thrombocytopenia	9 (10.0)	14 (15.6)	22 (12.0)	5 (2.7)
Non-hematological (%)				
Respiratory system	20 (22.2)	13 (14.4)	22 (12.0)	9 (4.9)
Gastrointestinal tract	23 (25.6)	5 (5.5)	12 (6.6)	4 (2.2)
Renal insufficiency	2 (2.2)	2 (2.2)	0 (0)	1 (0.5)

dCRT: definitive chemoradiotherapy.

## Data Availability

Data presented in this study will be provided upon reasonable request.

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
