# Peer review of "Treatment Outcomes and Risk Factors for Incomplete Treatment after Definitive Chemoradiotherapy for Non-Resectable or Metastatic Esophageal Cancer"

_cancers, 2023, doi:10.3390/cancers15225421_

Round 1

Reviewer 1 Report

Comments and Suggestions for Authors

General comment:

The authors analyzed the prognostic factors for patients with non-resectable or metastatic esophageal cancer treated by chemoradiotherapy. The results revealed that Incom-35 plete definitive treatment is associated with far worse prognosis. Poor performance, bone metastasis, airway invasion, and liver cirrhosis are risk factors for incomplete treatment. The findings are interesting and clinically useful. However, there are some points that the authors should clarify.

Specific comment:

1. The radiation dose is misleading. cGy? or Gy? (as shown in Table 1), the authors should clarify this.

2. How do the authors define "treatment imcompletion"? In table 1, there are 34.4% of patients in the "treatment incomplete" group actually received radiation dose greater than 5040cGy, which is above the so called standard dose for esophageal cancer.

3. As radiation side effects increase with the dose irradiated, did the authors analyzed how much was the dose that patients in the incompletion group actually received?

4. The authors cited the reference of ARTDECO study to discuss about the dose needed. However, this study has been criticized that the mean age was over 70 years. Besides, the high dose group in the study actually received large daily dose to the esophagus which can lead unnecessary radiation reactions. The authors should discuss about these points.

Reviewer 2 Report

Comments and Suggestions for Authors

Pai, et al, report a manuscript of clinical data of treatment outcomes and risk factors for imcomlete treatment after definitive chemoradiotherapy for unresectable or metastatic esophageal cancer in one center. There are some points need to be clarified.

1.      It would be better to mention that the experience from “single medical center” in your topic

2.      In materials and methods section, the IRB was approved in 2015. However, your patients sources are from Jan 2010 to Dec 2019. It is not “prospective” study. If you used databast from other department, like “cancer cetner”, please mention it in your materials and methods section. If not, you may need to provide other IRB number (before Jan 2010, you collect data).

3.      Table 1, you defined definitive CCRT as R/T cumlative dose of 50.4Gy or above (line 99). However, in your complete dCCRT group, l66 (36.1%) patients received equal or less than 5040 Gy and 31 (34.4%) patients received more than 5040 Gy in incomplete dCCRT group. It is not compatible with your defination.

4.      All patients received time series follow up. Please defined the response in material or methods section. Do you evaulate after dCCRT or 3 months, 6 months, 1 year, 2year………after treatment? Please fill clinical response to therapy in incomplete dCCRT group, in Table 1

5.      What does MDT conference mean in Table 1? Please make it clear.

6.      Please add follow up period of each groups in table 1. That would let the data more clear (patients can not evaulate treatment response is due to lose of follow up or death before response evaluation date?)

7.      Although we know that treatment response would influence on OS, it would be better you put response in your univariable and multivariable analysis (Table 2)

8.      The survival rate is not only related to clinical factors, but also treatment choice. Chemotherapy choice is important. Please mention how many percentage of patients received single platinum, single 5FU, combination, or single Taxane/combination with platinum target therapy (eg, cetuximab), anti-PD1 antabonist and others, in each group if possible.

9.      How many patients received salvage chemotherapy/immunotherapy in each group? It will be influence on survival.

10.  When did you arrange salvage esophagectomy? How long after dCCRT? How many patients in gropu 3 received further systemic treatment after salvage esophagectomy?

Reviewer 3 Report

Comments and Suggestions for Authors

Manuscript entitled "Treatment outcomes and risk factors for incomplete treatment after definitive chemoradiotherapy for non-resectable or metastatic esophageal cancer"

Major issues:

1. The parmeter included for analysis is limited. The aithors should include pathologic factors as well. They should include tumor thickness, depth of invasion, vascular invasion, differentiation, ... etc.

2. The survival curves of important paraneters hould be disclosed.

3. The correlation between important parameters should be analyzed.

4. More survival enpoints should be included.

Comments on the Quality of English Language

English polishing is mandatory.

Reviewer 4 Report

Comments and Suggestions for Authors

Comments to the authors:

In this study, the authors investigated the treatment outcomes and risk factors for incomplete treatment after definitive chemoradiotherapy for non-resectable or metastatic esophageal cancer. The authors provided some critical data that may be helpful for clinical therapy. I have some concerns that need to be addressed by the authors before publication. See below:

1: Lack of Context: The abstract does not provide any context for the study, such as the prevalence and significance of esophageal cancer, making it difficult for readers to understand the broader implications of the research.

2: Incomplete Information: The abstract mentions "treatment-related adverse events" without specifying what these events are, which is a critical aspect of the study. Readers need to know the nature and severity of these adverse events to assess the actual impact on patients.

3: Limited Generalizability: The study appears to focus on a specific population of patients treated at a single hospital between 2010 and 2019, which may limit the generalizability of the findings to a broader population or more recent advancements in treatment.

4: Complex Data: The manuscript includes a lot of statistical and technical information, which might make it challenging for a general audience to grasp the key findings and their implications. A more simplified and reader-friendly presentation would be beneficial.

5: Statistical Jargon: The abstract includes numerous statistical terms like "hazard ratio," "multivariable analysis," and "Cox proportional hazards regression," which may be difficult for non-specialists to understand. Providing some plain-language interpretations or definitions of these terms would make the abstract more accessible.

Inadequate Discussion of Methodology: The methodology section is sparse and lacks details about how data were collected, patient selection criteria, and the specific treatments administered. Providing more information about the study design and methodology would enhance the rigor and transparency of the research.

6: No Mention of Limitations: The abstract should discuss the limitations of the study, such as potential biases, limitations in the data, or any confounding factors. This would help readers understand the boundaries of the research.

7: Imprecise Language: The manuscript uses imprecise language in some places, such as stating that "complete treatment was related to increased survival" without specifying the degree of improvement. Precision in reporting results is essential.

8: The following paper should be cited: Movahedi , F., Li, L., & Xu, Z. (2023). Repurposing anti-parasite benzimidazole drugs as selective anti-cancer chemotherapeutics. Cancer Insight2(1), 31–52. https://doi.org/10.58567/ci02010003

To improve the manuscript, the authors should consider addressing these issues, providing context, simplifying language, and highlighting the broader implications of their research. Additionally, including a clear, concise conclusion would help readers understand the key takeaways from the study.

Comments on the Quality of English Language

Moderate editing of English language required

Round 2

Reviewer 3 Report

Comments and Suggestions for Authors

The revision is acceptable for publication.

Comments on the Quality of English Language

Acceptable

Reviewer 4 Report

Comments and Suggestions for Authors

The authors have addressed my concerns.